# Vesicular Transport of Encapsulated microRNA between Glial and Neuronal Cells

**DOI:** 10.3390/ijms21145078

**Published:** 2020-07-18

**Authors:** Walter J. Lukiw, Aileen I. Pogue

**Affiliations:** 1Departments of Neurology, Neuroscience and Ophthalmology, Neuroscience Center and Department of Ophthalmology, Louisiana State University Health Sciences Center, New Orleans, LA 70112, USA; 2LSU Neuroscience Center, Louisiana State University Health Sciences Center, New Orleans, LA 70112, USA; 3Department of Ophthalmology, LSU Health Sciences Center, New Orleans, LA 70112, USA; 4Department of Neurology, Louisiana State University Health Sciences Center, New Orleans, LA 70112, USA; 5Alchem Biotech Research, Toronto, ON M5S1A8, Canada; apogue@yahoo.com

**Keywords:** Alzheimer’s disease, extracellular microvesicles (EMV), exosomes (EX), microRNA-146a

## Abstract

Exosomes (EXs) and extracellular microvesicles (EMVs) represent a diverse assortment of plasma membrane-derived nanovesicles, 30–1000 nm in diameter, released by all cell lineages of the central nervous system (CNS). They are examples of a very active and dynamic form of extracellular communication and the conveyance of biological information transfer essential to maintain homeostatic neurological functions and contain complex molecular cargoes representative of the cytoplasm of their cells of origin. These molecular cargoes include various mixtures of proteins, lipids, proteolipids, cytokines, chemokines, carbohydrates, microRNAs (miRNA) and messenger RNAs (mRNA) and other components, including end-stage neurotoxic and pathogenic metabolic products, such as amyloid beta (Aβ) peptides. Brain microglia, for example, respond to both acute CNS injuries and degenerative diseases with complex reactions via the induction of a pro-inflammatory phenotype, and secrete EXs and EMVs enriched in selective pathogenic microRNAs (miRNAs) such as miRNA-34a, miRNA-125b, miRNA-146a, miRNA-155, and others that are known to promote neuro-inflammation, induce complement activation, disrupt innate–immune signaling and deregulate the expression of neuron-specific phosphoproteins involved in neurotropism and synaptic signaling. This communication will review our current understanding of the trafficking of miRNA-containing EXs and EMVs from astrocytes and “activated pro-inflammatory” microglia to target neurons in neurodegenerative diseases with an emphasis on Alzheimer’s disease wherever possible.

## 1. Microvesicle Signaling in Neurodegeneration

### Overview

#### Exosomes (EX) and Extracellular Microvesicles (EMV)

The exosome (EX) and extracellular microvesicle (EMV), sometimes referred to collectively as extracellular microparticles (MP) or extracellular organelles, are lipid-bilayer membrane encapsulated, nanosized particles 30–1000 nm in diameter that play essential roles: **(i)** in intercellular communication and tissue homeostasis by transporting diverse classes of biologically active molecules between cells of glial and neuronal origins (and probably vice versa; glial cells include microglial cells, astroglia and astrocytes); **(ii)** in maintaining the crosstalk between neurons and glia and neural cells with the peripheral nervous system (PNS) and systemic circulation; **(iii)** in synaptic plasticity, neuroprotection, neuroregeneration, maintenance of the neuronal–glial interface, and the dissemination of potentially neuropathological molecules (Colombo et al., 2014 [1]; Choi et al., 2015 [2]; Tabata 2015 [3]; Xu et al., 2018 [4]; Stahl et al., 2019 [5]; Arbo et al., 2020 [6]; Barnes and Somerville 2020 [7]; Birgisdottir and Johansen 2020 [8]; Song et al., 2020 [9]). As such, each EX or EMV carry their own lipidome, proteome, lipoproteome, and/or transcriptome as an individually packaged organelle destined for independent extracellular targeting. Other classes of much larger apoptosis-related extracellular bodies 1–5 μm in diameter, sometimes called “apoptotic bodies”, involved in brain cell and tissue degradation have been recently reviewed in depth and will not be dealt with further in this communication (van Niel et al., 2018 [10]; Stahl et al., 2019 [5]; Rodriguez-Quijada and Dahl 2020 [11]).

Exosomes (EXs): **(i)** were first characterized about ~40 years ago as a heterogeneous group of cell cytoplasm-derived intracellular micro-particles (MPs) of endosomal origin (also known as ectosomes, shedding vesicles, microvesicles or EMVs); **(ii)** are variable in their content and morphology, ranging in size from about ~30 to ~100 nm in diameter; **(iii)** are released by a variety of central nervous system (CNS) cells whose function was originally described as a cytological basis for intercellular signaling and communication (Trams et al. 1981 [12]; Columbo et al., 2014 [13]; Jiang et al., 2019 [14]; Mathews and Levy 2019 [15]; Arbo et al., 2020 [6]; Martins et al., 2020 [16]). EXs originated as intracellular endosomes containing intraluminal vesicles (ILV) that fuse with the plasma membrane and empty their plasma membrane-encapsulated cargo. ILVs are released once extracellular bodies become exosomes (Mathews and Levy 2019 [15]; Arbo et al., 2020 [6]). Extracellular microvesicles (EMVs), 100–1000 nm in diameter, are lipid-bilayer membrane bound spheres and are typically released from the exterior plasma membrane of astrocytes and activated microglia, directly from the outward blebbing of the plasma membrane, and carry intracellular contents from their cells of origin. These include various complex mixtures of proteins, lipids, proteolipids, carbohydrates, miRNAs, mRNAs, cytokines, chemokines and end-stage metabolic products, including neurotoxic 42 amino acid amyloid-beta (Aβ42) peptides. In fact, the beta amyloid precursor protein (βAPP)-processing enzymes, gamma- and beta-secretase, that give rise to Aβ42 peptides have been found in abundance in both EXs and EMVs, and may represent part of an Aβ42 peptide clearance mechanism throughout the brain (Arbo et al., 2020 [6]; Webers et al., 2020 [17]). Similarly, the accrual in EXs of the trans-active response DNA-binding proteolipid of 43 kDa (TDP-43), a neurotoxic and pathological hallmark for amyotrophic lateral sclerosis (ALS) and frontotemporal degeneration (FTD), may be part of a critical shuttling pathway for the clearance of TDP-43 from diseased brain cells (Iguchi et al., 2016) [18]. Astrocyte-derived EMVs, in addition, have been shown to encapsulate and subsequently release various neuroprotective and neurorestorative factors, including fibroblast growth factor-2, vascular endothelial growth factor (VEGF), apolipoprotein-D, heat shock proteins (HSP), synapsin-1 (SYN1) and glutamate transporters, to their extracellular targets which include neurons (Stahl et al., 2019 [5]; Leidal and Debnath 2020 [19]; Upadhya et al., 2020 [20]). Some of these EMV cargoes, such as end-stage oxidized metabolic products and excesses of cytokines and chemokines, may be pathogenic and promote the spread of pro-inflammatory signaling and inflammatory neurodegeneration (Prada et al., 2018 [21]; Serpente et al., 2020 [22]; Vanherle et al., 2020 [4]; Figure 1). Pharmacologically-based intervention strategies restraining the release of EMVs by activated astrocytes, activated microglia or modulating their cargoes may be a useful therapeutic approach and beneficial for treating inflammatory neurodegenerative diseases (Mao et al., 2015 [23]; Mathieu et al., 2019 [24]; Arbo et al., 2020 [6]; Seyedrazizadeh et al., 2020 [25]; Upadhya et al., 2020 [20]). As most EMVs pass freely across the blood–brain barrier into systemic circulation, the detection of astrocyte-specific biomarkers in different neurological conditions may be additionally useful for monitoring disease progression, remission and/or therapeutic efficacy (Mao et al., 2015 [23]; Urbanelli et al., 2016 [26]; Federici et al., 2020 [27]; Groot and Lee 2020 [28]; Stahl et al., 2019 [5]; Upadhya et al., 2020 [20]; Vanherle et al., 2020 [4]).

Interestingly, EX, MP and EMV releases are common features of many diverse cell types in many different kinds of organisms, from prokaryotes to eukaryotes, and their formation and release have been widely documented in the gastrointestinal (GI) tract bacteria such as *Escherichia coli* and *Bacteroides fragilis*, species of the family *Brassicaceae* such as *Arabidopsis thaliana*, Protists, protozoa such as *Amoeba proteus* and slime molds such as *Dictyostelium discoideum*, nematodes such as *Caenorhabditis*, and up the evolutionary scale to a wide range of mammals including humans. The biogenesis, secretion and the release of EXs, MPs and/or EMVs into the extracellular space or external environment for the purpose of mediating intercellular communication or transmitting DNA- or RNA-encoded genetic information between different cell types and the environment is therefore a very ancient and conserved evolutionary process (Columbo et al., 2014 [13]; Arbo et al., 2020 [6]). From what is currently known, the EXs, MPs and/or EMVs of all prokaryotes and eukaryotes examined so far all carry small non-coding RNAs (sncRNAs), microRNA-like and/or microRNAs, as a fundamental system for the extracellular exchange and/or transmission of genetic information and there is the intriguing potential here for interspecies signaling and communication (Pogue et al. 2014 [33]; Cong et al., 2018 [34]; Avsar et al., 2020 [35]).

EXs and EMVs: **(i)** are released under normal physiological conditions, but are also discharged from parent cells upon cellular activation, hypoxia and/or hyperoxia, senescence, apoptosis and disease via a paracrine- and endocrine-type type action to their target cells; **(ii)** represent one of the major biological mechanisms for genetic exchange, immune signaling and the spread of inflammation and disease between cells of the host; **(iii)** EX and EMV trafficking in the mammalian CNS is a particularly robust, active and dynamic process (Valadi et al., 2007 [26]; Hunter et al., 2008 [36]; Deng et al., 2018 [34]; Stahl et al., 2019 [9]; Arbo et al., 2020 [6]; Hou et al., 2020 [37]). The human brain and retina, CNS, cerebrospinal fluid (CSF), neurovasculature, as well as the systemic circulation, are particularly rich sources of EXs, MPs and EMVs, suggesting that they are significant components of a highly active multi-component extracellular signaling and communication system (Trotta et al., 2018 [38]; Seyedrazizadeh et al., 2020 [39]; Song et al., 2020 [40]). Indeed, the magnitude of EX and ECV cargoes and the complexity of their content of multiple mediators make them a more formidable form of intercellular communication than the transfer of individual molecules, and the extracellular transfer of EX- or ECV-derived vesicular cargoes to neurons and other cell types may in fact permanently alter or modulate the phenotype of target cells (Stahl et al., 2019 [9]; Vanherle et al., 2020 [4]).

## 2. Extracellular Trafficking of EX and EMV Cargo

The biogenesis of EXs and EMVs, and the triggers for their release from parent cells, are complex biological processes that eventually require the navigation of these vesicles through the extracellular matrix to their extracellular targets which may be near or distant (Barnes and Somerville 2020 [7]; Birgisdottir and Johansen 2020 [41]; BriteS 2020 [42]; Jadli et al., 2020 [43]). Protein interaction maps have indicated that EX and EMV biolipid boundary membranes have a tendency to interact with discrete extracellular “node” proteins involving surface ligands and delivery receptors, indicating an “*extracellular directional strategy*” for vesicular sorting and translocation over both short and long distances (Choi et al., 2015 [44]; Leidal and Debnath 2020 [45]). Many of these directional pathways and sorting mechanisms remain incompletely understood. EX and EMV targets may be either locally or distally located within the CNS or, after the passage of vesicles from the CNS into the systemic circulation, vesicular cargoes may target extra-neural cell receptors throughout the CNS (Barbagallo et al., 2020 [46]; Serpente et al., 2020 [11]).

The dynamics and observed variability of EXs and EMVs in vesicle size, morphology and cargo further indicates: **(i)** that these vesicular organelles consist of a unique repertoire of cytoplasmic components representing cellular, molecular and genetic information that is a direct reflection of the unique biological condition of the parent cell’s cytoplasm at the time of vesicular release; **(ii)** that these microparticles play important roles as enveloped proteolipids, a nucleic acid-enriched “*information packet*” in a complex extracellular communication network; **(iii)** that EXs and EMVs may reprogram recipient, adjacent cells and/or distant tissues as CNS-resident cells involved in immune-surveillance and the maintenance of normal cellular homeostasis, while also contributing to neuropathology during disease; **(iv)** that the molecular content and rates of production and secretion of EXs and EMVs vary greatly depending on the cell-type and physiological state of the cells of vesicular origin (van Niel et al., 2018 [47]; Mathieu et al., 2019 [15]; Brites 2020 [42]; Leidal and Debnath 2020 [45]). Attesting to the importance of EXs and EMVs in neurocellular and CNS functions, it is remarkable that the genetic and pharmacological inhibition or content modification of EX or EMV secretion from astrocytes has been found to play a role in the induction of amyloidogenesis, inflammation status and disease progression in several transgenic models of murine neurodegeneration, including the aging, amyloid-overexpressing 5xFAD mouse model (Dinkins et al., 2016 [48]; Barnes and Somerville 2020 [7]; Leidal and Debnath 2020 [45]).

## 3. Selective microRNAs in EXs and EMVs

MicroRNAs (miRNAs) are soluble, amphipathic, single-stranded non-coding RNAs (sncRNAs) 18- to 25-ribonucleotides (nt) in length whose RNA sequences have been very highly selected. Some miRNAs, such as the neurologically relevant miRNA-378, are very highly conserved, and their core ribonucleotide sequence has remained virtually unchanged in plants (*Atropa belladonna*) and higher animals (*Homo sapiens*) over many billions of years of evolution (*Arabidopsis thaliana–Homo sapiens* divergence ~ 1.5 billion years; Pogue et al., 2014 [32]; Cong et al., 2018 [1]; Zhao et al., 2018 [49]; Avsar et al., 2020 [35]). Abundant data now indicates that miRNAs: **(i)** are the smallest known gene information-carrying nucleic acids yet discovered; **(ii)** are important posttranscriptional and epigenetic regulators of mRNA abundance, speciation and complexity in aging, development, neurological health and disease processes; **(iii)** play pivotal roles in the initiation, development and propagation of many human CNS disorders including progressive terminal cancers and lethal, age-related neurological syndromes with an inflammatory component; **(iv)** are loaded into both EXs and EMVs and are a typical component of their vesicular cargoes(Lukiw et al., 1990 [50]; Lukiw 2012a [51]; Lukiw 2012b [52]; Zhao et al., 2015 [53]; Hosaka et al., 2019 [54]; Slota et al., 2019 [55]; Swarbrick et al., 2019 [5]; Barbagallo et al., 2020 [46]; Briand et al., 2020 [8]; Li et al., 2020 [19]).

To date about 2650 microRNAs have been characterized in *Homo sapiens* while only about 25–30 individual miRNA species, or fewer, are abundant and easily detected in the human brain and retina (Clement et al., 2016 [2]; Hammond 2016 [28]; manuscript in preparation). Interestingly, a small subgroup of at least seven brain- and retina-enriched miRNAs—including the miRNA-7/Let-7a cluster, miRNA-9, the miRNA-23a/27a cluster, miRNA-34a, miRNA-125b, miRNA-146a, miRNA-155 and others—have been shown to be inducible via the pro-inflammatory transcription factor NF-kB (p50/p65) complex due to the presence of multiple high affinity NF-kB (p50/p65) recognition features in the majority of their immediate promoters (Lukiw 2012a [51]; Lukiw 2012b [52]; Hammond 2016 [28]). The up-regulation of these seven miRNAs can account for the down-regulation of the expression of many AD-relevant genes, including complement factor H (CFH), the interleukin-1 receptor associated kinase-4 (IRAK-4), the ubiquitin conjugating enzyme E2A (UBE2A), the signal transduction “Wingless and Int-1” Wnt signaling complex, the triggering receptor expressed in myeloid cells (TREM2), the inhibitor-of-NF-kB signaling protein complex (IkBKG), the synaptic proteins synapsin-1 and synapsin-2 (SYN1 and SYN-2), 15-lipoxygenase (15-LOX), the vitamin D receptor (VDR), the 12-pass integral membrane spanning tetraspanin-12 protein complex (TSPAN-12) and several others that have highly interactive roles in innate immunity, cytokine (IL-1β) signaling, the clearance of end-stage metabolic products, synaptic failure, neurotropism and amyloidogenesis, and an up-regulation of NF-kB (p50/p650) regulated gene expression (Lukiw et al., 1990 [50]; Lukiw 2012a [51]; Lukiw 2012b [52]; Zhao et al., 2016 [56]; Wang et al., 2019 [57]; summarized in Figure 1 in Pogue and Lukiw 2018 [33]). Importantly, NF-kB (p50/p65) is rapidly induced by a large number of reactive-oxygen species (ROS)-inducing neurochemical and biophysical signals inducing those associated with viral and bacterial infections, ionizing radiation, cytokines and chemokines, Aβ42 peptides and hypoxia, and the neurogenetic coupling of the ROS-sensitive NF-kB (p50/p65) to the up-regulation of these NF-kB (p50/p65) sensitive miRNAs is a very tightly coupled and rapid genetic event (Hill et al., 2015 [30]; Alexandrov et al., 2019 [58]; Zhao and Lukiw 2018 [59]).

The discovery of the important role of miRNAs in the regulation of the transcriptome of a cell was made about ~15 years ago and was shortly followed by the first reports of altered miRNA abundance, speciation and complexity in the limbic system of Alzheimer’s disease (AD) brains (Lukiw 2007 [60]) This included a significant up-regulation of what are now known as “*pro-inflammatory miRNAs*”, including miRNA-34a, miRNA-125b, miRNA-146a, miRNA-155 and others in the parenchyma of the temporal lobe neocortex (Lukiw 2007 [60]; Valadi et al., 2007 [26]; Hunter et al., 2008 [36]; Sethi and Lukiw 2009 [22]; Zhao et al., 2015 [61]; Hammond 2016 [28]). The first reports of EXs and EMVs being loaded with ribonucleic acid cargoes, such as specific miRNAs and mRNAs, first came about ~12 years ago: **(i)** from the microarray analysis of mast cells involved in innate and adaptive immunity, autoimmunity, and inflammation (Valadi et al., 2007 [26]); **(ii)** from studies of specific miRNAs, mRNAs and angiogenic proteins in glioblastoma and neuroblastoma tumor cells that have released EXs and EMVs (Skog et al., 2008 [25]; Columbo et al., 2014 [13]; Briand et al., 2020 [8]). These extruded vesicles were subsequently observed to be taken up by normal host brain microvascular endothelial cells, thereby inducing carcinogenic type phenotypic change in these brain cells, stimulating tumor invasion, proliferation and cancer spread mediated by endothelial cells of the neurovasculature (Hunter et al., 2008 [36]; Skog et al., 2008 [25]). Very recently, evidence has been provided showing that EXs and EMVs derived from miRNA-containing natural killer (NK) cells contribute to immunological surveillance and their provision of specific miRNAs may be the first-line of defense in the regulation of tumor cell growth, cytotoxicity and metastasis diffusion in the CNS (Briand et al., 2020 [8]; Federici et al., 2020 [62]). As discussed more fully below, it has recently been shown that brain EXs and EMVs released from diseased astrocytes and *“activated microglia”* carry specific miRNA-enriched cargoes, enriched in miRNA-34a and miRNA-125b for example, that contribute to neuropathological spreading and the exacerbation of neuropathological processes in AD, ALS, Parkinson’s disease (PD), Huntington’s disease (HD), stroke and other neuro-inflammatory degenerative conditions. This indicates a definitive role for EX and EMV miRNA cargoes in neurological disease processes with an inflammatory component which may have considerable diagnostic, prognostic and/or therapeutic value (Prada et al., 2018 [63]; Mathieu et al., 2019 [15]; Ghaffari et al., 2020 [27]; Li et al., 2020 [19]; Serpente et al., 2020 [11]; Upadhya et al., 2020 [64]). Because brain cells can release EXs and EMVs which can pass from the brain and CNS into the blood, brain-derived vesicles isolated from the systemic circulation may be of use to monitor diseases operating in the CNS, thus improving clinical diagnoses and prognoses (Barbagallo et al., 2020 [46]). This may also be of use in analyzing the efficacy of pharmaceuticals and/or therapeutic interventions being used to intervene in the AD process via the analysis of EXs and EMVs in blood plasma (Cha et al., 2019 [65]; Barbagallo et al., 2020 [46]).

## 4. EMV and miRNA Cargoes in the Spreading of Inflammatory Neurodegeneration

Accumulating evidence continues to implicate secreted miRNAs, including EX and EMV-encapsulated miRNAs, in the pathogenic spreading of progressive, age-related and incapacitating neurodegenerative diseases with an uncontrolled or deregulated inflammatory component and synaptic deficits. These include Alzheimer’s disease (AD), age-related macular degeneration (AMD), Parkinson’s disease (PD), Huntington’s disease (HD), prion disease (PrD), multiple sclerosis (MS), Japanese and viral-induced encephalitis and many related amyloidopathies, tauopathies and synucleinopathies (Mcachlan et al., 1988 [24]; Zhao et al., 2016 [66]; Palpagama et al., 2019 [67]; Slota and Booth 2019 [55]; Wang et al., 2019 [57]; Fan et al., 2020 [68]; Groot and Lee, 2020 [69]; Lukiw 2020 [70]). From extensive research reports, it seems most likely that, in inflammatory neurodegenerative disease, the up-regulation of single miRNAs is much less important than increases in small families of pathological miRNAs in driving disease progression (Zhao et al., 2015 [61]; Lukiw 2020 [70]). While the alterations in the abundance of select miRNAs may be disease-specific, others such as miRNA-125b, miRNA-146a and miRNA-155 appear to be up-regulated during many of these neurological disorders and play highly integrated pathological roles. For example, an increase of miRNA-146a and miRNA-155 is observed in AD and AMD. Both miRNA-146a and miRNA-155 target the 3′ untranslated region (3′-UTR) of the complement factor H (CFH) mRNA in overlapping binding sites leading to a highly effective down-regulation in the expression of CFH and runaway loss-of-complement control, markedly increased pro-inflammatory signaling thereby contributing mechanistically to both AD and AMD inflammatory neuropathology (Hill et al., 2015 [30]; Fan et al., 2020 [68]). 

Importantly, the modification of miRNA expression to more homeostatic levels via anti-miRNA or anti-NF-kB strategies may be a useful therapeutic strategy to successfully address multiple aspects of neuropathological inflammation (Hammond 2015 [28]; Zhao et al., 2016 [66]; Ghaffari et al., 2020 [27]). CNS-derived EXs and EMVs that contain NF-kB-sensitive, pro-inflammatory miRNAs and other molecular biomarkers from the cells of their parental origin are known to cross the blood–brain barrier into the systemic circulation and may be reflective of the biochemical status for various neurodegenerative diseases. There are practical challenges associated with the methodology of the extraction and characterization of CNS-derived blood, CSF and tissue EXs and EMVs, and subsequent analysis of their miRNAs and other intra-vesicular cargoes, however the methodologies for EX and EMV isolation and categorization are constantly improving (Hornung et al., 2020 [71]; Serpente et al., 2020 [11]).

As previously discussed, EXs or EMVs and their miRNA cargoes may be shed into the extracellular environment under physiologically homeostatic or pathological conditions, either constitutively, or upon activation via acute injury (Kandiyil et al., 2019 [72]), hypoxia, hypoxia-induced oxidative stress and the generation of reactive oxygen species (ROS) by multiple mechanisms (Deng et al., 2018 [34]), by senescence or “*lingering death*” (Urbanelli et al., 2016 [20]), by pro-inflammatory mediators (including inflammatory cytokines, chemokines and Aβ peptides), by factors in the systemic circulation such as thrombin and thrombotic factors (Hunter et al., 2019 [73]), by components of the purinergic pathways (Ludwig et al., 2020 [74]) and microbial or viral virulence factors, including bacterial exotoxins and lipopolysaccharide (LPS; Mao et al., 2015 [75]; Lukiw et al., 2018 [76]; Alexandrov et al., 2019 [58]). As observed in cancer, inflammatory bowel diseases and neurodegenerative diseases, including AD and ALS, exosomes can carry adenosine receptors and other components of purinergic pathways, and the production and release of some EXs or EMVs may be induced by the stimulation of purinergic receptors by purine receptor agonists and reduced by purine receptor antagonists (Hosaka et al., 2019 [54]; Ludwig et al., 2020 [74]).

## 5. Unanswered Questions

Several important unanswered questions remain in our understanding of the generation and characterization of intraluminal vesicles (ILV), microparticles (MP), exosomes (EX) and extracellular microvesicles (EMV), the nature of their vesicular cargoes and miRNA composition. These include: **(i)** the signals and pathways essential for stimulation and the origin of their formation, as well as the mechanism of their release from many different cell types in the CNS and their proficiency for modulating functions of target cells; **(ii)** the molecular-genetic injury and/or environmental factors which stimulate their release for this evolutionarily-conserved type of information communication system amongst astrocytes, microglia and neurons; **(iii)** their increased production and release during the initiation and spread of progressive age-related inflammatory neurodegeneration; **(iv)** the actual molecular contents, stoichiometry and packaging of the contents in the vesicles themselves; **(v)** the magnitude and signaling impact of their plasma membrane-packaged vesicular cargo; **(vi)** the regulation of their trafficking and targeting to neuronal cells via plasma membrane-mediated cell-surface reception mechanisms; **(vii)** the lipidomic, proteomic and transcriptomic profiles of these vesicles and what miRNA and/or mRNA encoded information these vesicles may be carrying; **(viii)** whether or not EXs and/or EMVs can transfer their miRNA-enriched intraluminal cargoes to other cell types and/or to other species; **(ix)** the role of circular RNA (circRNA) which have been shown in some cases to act as natural *“anti-miRNA sponges*” of specific miRNA activities (Lukiw 2013 [77]; Zhao et al., 2016 [78]; Fakhoury 2018 [53]; Pogue and Lukiw., 2018 [33]; Zhao and Lukiw, 2018 [59]; Avsar et al., 2020 [35]; Groot and Lee 2020 [69]; Hou et al., 2020 [37]; Li et al., 2020 [19]; Ma et al., 2020 [79]; Serpente et al., 2020 [11]). A single microvesicle, for example, may carry a cargo consisting of a complex cocktail of multiple proteins, proteolipids, cytokines, chemokines, carbohydrates, miRNAs, mRNAs, circRNAs and/or other nucleic acid signals as well as neurotrophic and amyloidogenic factors together representing novel, unique and discrete “*information packets*” for the structural and functional support of target cells, and acting as unique mediators of intercellular information transfer. 

## 6. Concluding Remarks

The neurobiology of EX and EMV genesis, release, translocation and uptake by target cells, and their containment of select miRNA populations enriched in the CNS, indicate that they are significant components of a highly dynamic system of intercellular communication via extracellular translocation and targeting in brain cell health, aging and disease. Multiple independent studies indicate that while vesicle-mediated intercellular signaling is important in the homeostatic maintenance of brain cell functions, they have a substantial role in the proliferation of injury, cancer and inflammatory neurodegeneration signaling, as is observed in the AD brain. Many of the details of the mechanisms by which EXs and EMVs and their miRNA cargoes are generated and released by the activation of astrocytes and microglia and their trafficking to target brain cells, primarily neurons, remain to be further clarified. A greater understanding of the mechanisms underlying EX and EMV biogenesis, cargo selection and loading, vesicle release, translocation exterior to the cells of origin and targeting to adjacent or distant neural cells remains critical for unlocking the immense neurobiological and therapeutic potential for these ubiquitous organelles. Not only could an increased understanding of EX and EMV systems and their containment of miRNAs in the brain be useful in the treatment of CNS injuries and progressive age-related inflammatory neurodegeneration, but may also prove useful as delivery vehicles for therapeutic miRNAs, anti-miRNAs and both bioactive and neuroactive pharmaceuticals.

## Figures and Tables

**Figure 1 ijms-21-05078-f001:**
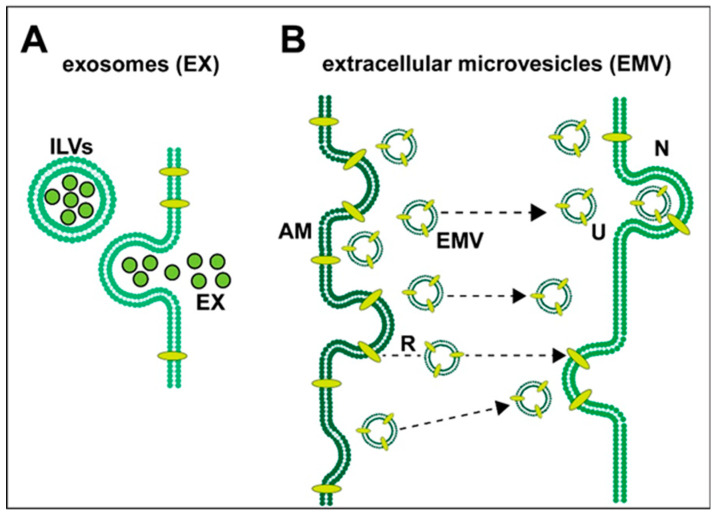
Highly schematized representation of (**A**) the release of exosomes (EXs) and (**B**) the release and uptake of extracellular microvesicles (EMVs) in neural cells of the human central nervous system (CNS); both types of vesicular transport systems have been observed to operate in the brain between astroglial cells and neurons; (**A**) exosomes—when mature intracellular endosomes (also known as multi-vesicular bodies) containing intraluminal vesicles (ILVs; black outlined green spheres) fuse with the plasma membrane and empty their plasma membrane-encapsulated cargo, ILVs are released and, from being extracellular, they become exosomes (EX); these 30–100 nm diameter spheres contain various mixtures of proteins, lipids, proteolipids, cytokines, chemokines, microRNAs (miRNA), messenger RNAs (mRNA) and end-stage neurotoxic metabolic products, including 42 amino acid amyloid-beta (Aβ42) peptides, tau proteins and/or the lipid raft associated flotillin (Angelopoulou et al. 2020 [29]; Hornung et al., 2020 [30]); EXs appear to play a central role in the spread of Aβ42 pathology and amyloidogenesis (Mathews and Levy 2019 [15]; Arbo et al., 2020 [6]; Peng et al., 2020 [31]); (**B**) extracellular microvesicles (EMVs)—the exterior plasma membrane of activated microglia (AM) can release (R) 100–1000 nm diameter EMVs directly from the outward blebbing of the plasma membrane of microglias and astrocytes and carry intracellular contents from their cells of origin; this includes various complex mixtures of proteins, lipids, proteolipids, miRNAs, mRNAs, end-stage metabolic products and cytokines and chemokines; together these EMV contents may be pathogenic and cause the spread of pro-inflammatory signaling and inflammatory neurodegeneration (Prada et al., 2018 [21]; Serpente et al., 2020 [22]; Vanherle et al., 2020 [4]). Microvesicular trafficking and the extracellular microvesicle uptake (U) by neurons (N) via directed translocation mechanisms (dashed black lines with black arrowheads) may occur via the direct fusion of the EMV membrane with the neuronal cell plasma membrane or by endocytosis (Stahl et al., 2019 [5]; Arbo et al., 2020 [6]; Hornung et al., 2020 [30]; Peng et al., 2020 [32]; Song et al., 2020 [9]; Upadhya et al., 2020 [20]).

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
