# Peer review of "Vesicular Transport of Encapsulated microRNA between Glial and Neuronal Cells"

_ijms, 2020, doi:10.3390/ijms21145078_

Round 1

Reviewer 1 Report

Authors reviewed the cell-cell interaction with exosomes (EX) and extracellular microvesicles (EMV) in the central nervous system, by introducing many papers in the field. They especially described microRNAs contained by those vesicles in detail. The paper was well-written, however there are some points for requirement or recommendation of revisions as below.

  1. Authors showed releasing manners of EX and EMV, and uptake manner of EMV in Fig, 1. It is recommended to add one more figure graphically showing communication roles of contents including microRNAs in EX and EMV between glial cells and neurons.
  2. The manuscript includes some typographical errors, for example “1-5 um” in line 10 of P. 2, and “idegradation” in line 11 of P. 2. LEGEND TO FIGURE 1 in P. 3 includes sentences in boldface type. The second paragraph in P. 5 includes an improper line break.
  3. A literature (Palpagama et al., 2019) is unsuitably referred in line 12 of P. 6. The paper does not include EX or EMV.

Author Response

IJMS Editorial Office

=================================================

REVIEWER #1: Comments and Suggestions for Authors

Authors reviewed the cell-cell interaction with exosomes (EX) and extracellular microvesicles (EMV) in the central nervous system, by introducing many papers in the field. They especially described microRNAs contained by those vesicles in detail. The paper was well-written, however there are some points for requirement or recommendation of revisions as below.

  1. Authors showed releasing manners of EX and EMV, and uptake manner of EMV in Fig, 1. It is recommended to add one more figure graphically showing communication roles of contents including microRNAs in EX and EMV between glial cells and neurons.
  2. The manuscript includes some typographical errors, for example “1-5 um” in line 10 of P. 2, and “idegradation” in line 11 of P. 2. LEGEND TO FIGURE 1 in P. 3 includes sentences in boldface type. The second paragraph in P. 5 includes an improper line break.
  3. A literature (Palpagama et al., 2019) is unsuitably referred in line 12 of P. 6. The paper does not include EX or EMV.

Submission Date 27 June 2020 Date of this review 08 Jul 2020 11:42:19

RESPONSE TO REVIEWER #1:

Firstly we would like to kindly and sincerely thank Reviewer #1 for their valuable time and expertise in the review of our work – the additions, corrections and clarifications to our manuscript have resulted in a significantly stronger contribution to IJMS – thanks again

  1. Thank you for the interesting comment but Reviewer #1’s suggestion to add a Figure is well beyond the scope of this paper – this is an invited review article is about ‘Vesicular transport of encapsulated microRNA between glial and neuronal cells’ and not what microRNAs do – ‘What microRNAs do’ is the subject of many recent papers (almost 5,000 references in the last 6 months on MedLine; www.ncbi.nlm.nih.gov); several of these most relevant publications are referenced in the current manuscript), including:

Brites D. Regulatory function of microRNAs in microglia. Glia. 2020;68(8):1631-1642. doi:10.1002/glia.23846

Fan W, Liang C, Ou M, Zou T, Sun F, Zhou H, Cui L MicroRNA-146a is a wide-reaching neuroinflammatory regulator and potential treatment target in neurological diseases Frontiers in Molecular Neuroscience https://doi.org/10.3389/fnmol.2020.00090 2020

Lukiw WJ microRNA-146a signaling in Alzheimer's disease (AD) and prion disease (PrD) Frontiers in Neurology  doi: 10.3389/fneur.2020.0046, in press 2020

Serpente M, Fenoglio C, D'Anca M, et al. MiRNA profiling in plasma neural-derived small extracellular vesicles from patients with Alzheimer's disease. Cells. 2020; 9(6):E1443. Published 2020 Jun 10. doi:10.3390/cells9061443

  1. All typographical errors have been corrected – thank you Reviewer #1 for pointing these out;
  2. Thank you again for pointing this reference error – the ’Palpagama et al., 2019’ reference has been changed to (2 references):

Mathieu, L. Martin-Jaular, G. Lavieu, C. Thery Specificities of secretion and uptake of exosomes and other extracellular vesicles for cell-to-cell communication Nat. Cell Biol., 21 (1) (2019), pp. 9-17

and

Prada I, Gabrielli M, Turola E, et al. Glia-to-neuron transfer of miRNAs via extracellular vesicles: a new mechanism underlying inflammation-induced synaptic alterations. Acta Neuropathol. 2018;135(4):529-550. doi:10.1007/s00401-017-1803-x

=======================================

END OF RESPONSES

Lastly, we would again like to thank the Reviewers for their valuable time and expertise in the review of our work. The incorporation of their suggestions and the addition of 3 very recent references from the last 18 months (2019, 2020) have added greatly to the suitability of the publication of this work in IJMS – thanks again

Sincerely

 ==============================================r J. Lukiw BS, MS, PhD, Professor of Neurology, Neuroscience and Ophthalmology, Bollinger Professor of Alzheimer’s disease (AD), LSU Neuroscience Center, Louisiana State University Health Sciences Center, 2020 Gravier Street, Suite 904, New Orleans LA 70112 USA

TEL 504-599-0842 EMAIL wlukiw@lsuhsc.edu

Reviewer 2 Report

In this review, authors discuss vesicular transport system (EXs and EMVs) in neural cells, especially from atrocytes and microglia to neurons. This review provides interesting information, however there is a suggestion on this manuscript.

It would be  desirable if authors add table on features and roles of EXs and EMVs in neural cells under physiological and pathological conditions.

Author Response

REVIEWER #2: Comments and Suggestions for Authors

  1. In this review, authors discuss vesicular transport system (EXs and EMVs) in neural cells, especially from atrocytes and microglia to neurons. This review provides interesting information, however there is a suggestion on this manuscript.

It would be desirable if authors add table on features and roles of EXs and EMVs in neural cells under physiological and pathological conditions.

Submission Date 27 June 2020 Date of this review 06 Jul 2020 04:05:00

RESPONSE TO REVIEW:

Firstly, we would like to kindly and sincerely thank Reviewer #2 for their valuable time and expertise in the reading and critical review of our current manuscript – the additions, corrections and clarifications to our manuscript have resulted in a significantly stronger contribution to IJMS – thank you

  1. Multiple tables on features and roles of EXs and EMVs in cells (including their comparison to ‘apoptotic bodies’) in health and have been provided in depth in very many recent reviews - and to avoid excessive duplication have not been (again) presented here – rather they have been directly referenced in the manuscript text – very comprehensive relevant references containing such tables – and/or highly descriptive figures - are present in the following recent publications:

Ståhl AL, Johansson K, Mossberg M, Kahn R, Karpman D. Exosomes and microvesicles in normal physiology, pathophysiology, and renal diseases. Pediatr Nephrol. 2019;34(1):11-30. doi:10.1007/s00467-017-3816-z

van Niel G, D'Angelo G, Raposo G. Shedding light on the cell biology of extracellular vesicles. Nat Rev Mol Cell Biol. 2018;19(4):213-228. doi:10.1038/nrm.2017.125

To further allay the concerns of Reviewer #2 three additional very recent publications on EXs and EMVs in Alzheimer’s disease and other disorders have been added in the revised manuscript text:

Groot M, Lee H. Sorting mechanisms for microRNAs into extracellular vesicles and their associated diseases. Cells. 2020; 9(4):1044. Published 2020 Apr 22. doi:10.3390/cells9041044

Jiang L, Dong H, Cao H, Ji X, Luan S, Liu J. Exosomes in pathogenesis, diagnosis, and treatment of Alzheimer's disease. Med Sci Monit. 2019;25:3329-3335. Published 2019 May 6. doi:10.12659/MSM.914027

Martins TS, Trindade D, Vaz M, et al. Diagnostic and therapeutic potential of exosomes in Alzheimer's disease [published online ahead of print, 2020 Jul 3]. J Neurochem. 2020;10.1111/jnc.15112. doi:10.1111/jnc.15112

Again, these 3 recent publications either present features and roles of EXs and EMVs in cells in very detailed tabular or diagrammatic form.

=======================================

END OF RESPONSES

 Lastly, we would again like to thank the Reviewers for their valuable time and expertise in the review of our work. The incorporation of their suggestions and the additions of 3 very recent references have added greatly to the suitability of this review publication in IJMS.

Sincerely

 ==============================================

Walter J. Lukiw BS, MS, PhD,Professor of Neurology, Neuroscience and Ophthalmology, Bollinger Professor of Alzheimer’s disease (AD), LSU Neuroscience Center, Louisiana State University Health Sciences Center, 2020 Gravier Street, Suite 904, New Orleans LA 70112 USA

TEL 504-599-0842, EMAIL wlukiw@lsuhsc.edu